# A Recurrent Neural Cascade-based Model for Continuous-Time Diffusion Process

## Abstract

Many works have been proposed in the literature to capture the dynamics of diffusion in networks. While some of them define graphical markovian models to extract temporal relationships between node infections in networks, others consider diffusion episodes as sequences of infections via recurrent neural models. In this paper we propose a model at the crossroads of these two extremes, which embeds the history of diffusion in infected nodes as hidden continuous states. Depending on the trajectory followed by the content before reaching a given node, the distribution of influence probabilities may vary. However, content trajectories are usually hidden in the data, which induces challenging learning problems. We propose a topological recurrent neural model which exhibits good experimental performances for diffusion modelling and prediction.

## 1 Introduction

The recent development of online social networks enabled researchers to suggest methods to explain and predict observations of diffusion across networks. Classical cascade models, which are at the heart of the research literature on information diffusion, regard the phenomenon of diffusion as an iterative process in which information transits from users to others in the network (Saito et al., 2008; Gomez-Rodriguez et al., 2011), by a so-called word-of-mouth phenomenon. In this setting, diffusion modeling corresponds to learning probability distributions of content transmission. Various cascade models have been proposed in the literature, each inducing its own learning process to explain some observed diffusion episodes and attempting to extract the main dynamics of the network. However, most of these models rely on a strong markovian assumption for which the probabilities of next infections[1] only depend on who is already infected at the current step, not on the past trajectories of the diffused content. We claim that the history of spread contains much valuable information that should be taken into account by the models. Past trajectories of the diffusion can give insights about the different natures of the contents. Also, the content may be changed during the diffusion, with different transformations depending on which nodes re-transmit the information.

On the other hand some recent approaches rely on representation learning and recurrent neural networks (RNN) to predict the future spread of diffusion given the past. A naive possibility would be to consider diffusion episodes as sequences of infections and propose temporal point process approaches to model the dynamics. Using the Recurrent Marked Temporal Point Process model (Du et al., 2016), the current hidden state of the RNN would embed the history of the whole diffusion sequence, which would be used to output the next infected node and its time of infection. However, since diffusion episodes are not sequences but trees, naive recurrent methods usually fail in capturing the true dynamics of the networks. Embedding the whole past in the state of a given node rather than restraining it to its specific ancestor branch leads to consider many independent and noisy events for future predictions. A model that would consider the true diffusion paths would be more effective, by focusing on the useful past. If the true diffusion paths were known, it would be possible to adapt works on recurrent neural models for tree structures such as successfully proposed in (Tai et al., 2015) for NLP tasks. Unfortunately, in most of applications the topology of diffusion is unknown while learning. In the task considered in this paper, the only observations available are the timestamps of the infected nodes.

---

[1] Throughout this paper, we refer to infection for denoting the participation of a node of the network in the diffusion.

To cope with it, (Wang et al., 2017a) proposed `Topo-LSTM`, a Long-Short Time Memory network that considers a known graph of relationships between nodes to compute hidden states and cells of infected nodes. The hidden state and cell of a given node at $t$ depend on those from each of its predecessors that have been infected before $t$. Since nodes may have multiple predecessors that are infected at time $t$, the classical `LSTM` cannot be applied directly. Instead, (Wang et al., 2017a) proposed a cell function that aggregates infector candidate states via mean pooling. This allows to take the topology of the possible diffusion into account, but not the past trajectory of the content (it averages all possible paths). To overcome this, (Wang et al., 2017b) proposed a cascade attention-based RNN, which defines a neural attention mechanism to assign weights to infector candidates before summing their contribution in the new hidden state at $t$. The attention network is supposed to learn to identify from whom comes the next infection based on past states. However, such an approach is likely to converge to most of attention weight vectors in the center of the simplex, since diffusion is a stochastic process with mostly very weak influence probabilities. The deterministic inference process of the approach limits its ability to produce relevant states by mixing from multiple candidates rather than sampling the past trajectories from their posterior probabilities. Note the similar approach in (Wang et al., 2018), which does not use a RNN but defines a composition module of the past via attention networks to embed the episode in a representation space from which the probability of the next infected node can be deduced. Beyond the limits discussed above w.r.t. deterministic mixing of diffusion trajectories, no delay of infection is considered in this work, which makes it impossible to use for diffusion modeling purposes.

Recently, many works in representation learning used random walks on graphs to sample trajectories that can be used to learn a network embedding which respects some topological constraints. While `DeepWalk` (Perozzi et al., 2014) only uses structural information, models proposed in (Nguyen et al., 2018) or (Shi et al., 2018) include temporal constraints in random walks to sample feasible trajectories w.r.t. observed diffusion episodes. The approach `DeepCas` from (Li et al., 2016) applies this idea for the prediction of diffusion cascades. However, such approaches require a graph of diffusion relations as input, which is not always available (and not always representative of the true diffusion channels of the considered network as reported in (Ver Steeg & Galstyan, 2013)). In our work, we consider that no graph is available beforehand. Moreover, no inference process is introduced in `DeepCas` to sample trajectories from their posterior probabilities given the observed diffusion sequences. The sampling of trajectories is performed in an initialization step, before learning the parameters of the diffusion model.

In this paper, we propose the first bayesian topological recurrent neural network for sequences with tree dependencies, which we apply for diffusion cascades modelling. Rather than building on a preliminary random walk process, the idea is to consider trajectory inference during learning, in order to converge to better representations of the infected nodes. Following the stochastic nature of diffusion, the model infers trajectories distributions from observations of infections, which are in turn used for the inference of infection probabilities in an iterative learning process. Our probabilistic model, based on the famous continuous-time independent cascade model (`CTIC`) (Saito et al., 2009) is able to extract full paths of diffusion from sequential observations of infections via black-box inference, which has multiple applications in the field. Our experiments validate the potential of the approach for modeling and prediction purposes.

The remaining of the paper is structured as follows. Section 2 presents some background and notations of the approach. Section 3 presents the proposed model. Section 4 reports experimental results of the approach compared to various baselines.

## 2 BACKGROUND

### 2.1 INFORMATION DIFFUSION

Information diffusion is observed as a set of diffusion episodes $\mathcal{D}$. Classically, episodes considered in this paper only contain the first infection event of each node (the earlier time a content reached the node). Let $\mathcal{U} = \{u_0, u_1, ...., u_{N-1}\}$ be a set of $N$ nodes, $u_0$ standing for the world node, used to model influences from external factors or spontaneous infections (as done in (Gruhl et al., 2004) for instance). A diffusion episode $D = (U^D, T^D)$ reports the diffusion of a given content in the network as a sequence of infected nodes $U^D = \left(U_0^D, ..., U_{|D|-1}^D\right)$ and a set $T^D = \{t_u^D \in \mathbb{N} + \infty | u \in \mathcal{U}\}$ of

infection time-stamps for all $u \in \mathcal{U}$. $U^D$ is ordered chronologically w.r.t. the infection time-stamps $T^D$. Thus, $U_i^D \in \mathcal{U}$ corresponds to the $i$-th infected node in $D$ for all $i \in \{0, ..., |D| - 1\}$, with $|D|$ the number of infected nodes in the diffusion. Every episode in $\mathcal{D}$ starts by the world node $u_0$ (i.e., $U_0^D = u_0$ for all episodes $D$). We note $t_u^D$ the infection time-stamp in $D$ for any node in $\mathcal{U}$, $\infty$ for nodes not infected in $D$. Time-stamps are relative w.r.t. to $t_1^D$, arbitrarily set to $t_1^D = 1$ in the data. Note that we also set $t_0^D = 0$ for every episode $D$. In the following, $D_i = (U_i^D, t_{U_i^D}^D)$ denotes the $i - th$ infected node in $U$ with its associated time-stamp.

Cascades are richer structures than diffusion episodes, since they describe how a given diffusion happened. The cascade structure stores the first transmission event $(u \to v)$ that succeeded from any node $u$ to each infected node $v$. Thus, a cascade $C^D = (U^D, T^D, I^D)$ corresponds to a transmission tree rooted in $u_0$ and reaching nodes in $U^D$ during the diffusion, according to a sequence $I^D$ of infector indices in $U^D$: for any $j \in \{1, ..., |D| - 1\}$ and any $i \in \{0, ..., |D| - 2\}$, $I_j^D$ equals $i$ iff $U_j^D$ was infected by $U_i^D$ in the diffusion $D$ (i.e., $U_i^D$ is the infector of $U_j^D$). We arbitrarily set $I_0^D = 0$ (no infector for the world node). Note that $I_1^D$ is always equal to 0, since there is no other candidate for being the infector of $U_1^D$ than the world $u_0$. For convenience, we note $I^D(i) \in U^D$ the infector of $U_i^D$ for $i \in \{0, ..., |D| - 1\}$. Note that the cascade structures respect that $t_{I^D(j)}^D < t_{U_j^D}^D$ for every $D$ and all $j \in \{1, ..., |D| - 1\}$ (the infector of a node $v$ is mandatory a node that was infected before $v$). Several different cascade structures are possible for a given diffusion episode. Cascade models usually perform assumptions on these latent diffusion structures to learn the diffusion parameters.

## 2.2 CASCADE MODELS

The Independent Cascade model (IC) (Goldenberg et al., 2001) considers the spread of diffusion as cascades of infections over the network. We focus in this paper on cascade models such as IC, which tend to reproduce realistic temporal diffusion dynamics on social networks (Guille et al., 2013). The classical version of IC is an iterative process in which, at each iteration, every newly infected node $u$ gets a unique chance to infect any other node $v$ of the network with some probability $\theta_{u,v}$. The process iterates while new infections occur. Since the expectation-maximization algorithm proposed in (Saito et al., 2008) to learn its parameters, IC has been at the heart of diffusion works.

However, in real life, diffusion occurs in continuous time, not discrete as assumed in IC. (Lamprier et al., 2016) proposed DAIC, a delay-agnostic version of IC, where diffusion between nodes is assumed to follow uniform delay distributions rather than occurring in successive discrete time-steps. A representation learning version of DAIC has been proposed in (Bourigault et al., 2016), where nodes are projected in a continuous space in a way that the distance between node representations render the probability that diffusion occurs between them. This allowed the authors to obtain a more compact and robust model than the former version of DAIC. Beyond uniform time delay distributions, two main works deal with continuous-time diffusion. NetRate (Gomez-Rodriguez et al., 2011) learns parametric time-dependent distributions to best fit with observed infection time-stamps. As NetRate, CTIC (Saito et al., 2009) uses exponential distributions to model delays of diffusion between nodes, but rather than a single parameter for each possible relationship, delays and influence factors are considered as separated parameters, which leads to more freedom w.r.t. to observed diffusion tendencies. Delays and influence parameters are learned conjointly by an EM-like algorithm. Note the continuous-time cascade model extension in (Zhang et al., 2018), which embeds users in a representation space so that their relative positions both respect some structural community properties and can be used to explain infection time-stamps of users.

In our model we consider that diffusion probabilities from any infected node $v$ depend on a latent state associated to $v$, which embeds the past trajectory of the diffused content. This state depends on the state of the node $u$ who first transmitted the content to $v$. Therefore, we need to rely on a continuous-time model such as CTIC (Saito et al., 2009), which serves as a basis for our work. In CTIC, two parameters are defined for each pair $(u, v)$ of nodes in the network: $k_{u,v} \in ]0; 1[$, which corresponds to the probability that node $u$ succeeds in infecting $v$, and $r_{u,v} > 0$, which corresponds to a time-delay parameter used in an exponential distribution when $u$ infects $v$. If $u$ succeeds in infecting $v$ in an episode $D$, $v$ is infected at time $t_v^D = t_u^D + \delta$, where $\delta \sim r_{u,v} \exp(-r_{u,v}\delta)$. These parameters are learned via maximizing the following likelihood on a set of episodes $\mathcal{D}$:

$$p(\mathcal{D}) = \prod_{D \in \mathcal{D}} p(D) = \prod_{D \in \mathcal{D}} \prod_{v \in U^D} h_v^D \prod_{v \notin U^D} g_v^D \tag{1}$$

where $h_v^D$ stands for the probability that $v$ is infected at $t_v^D$ by previous infected nodes in $D$ and $g_v^D$ is the probability that $v$ is not infected by any infected node in $D$.

We build on this in the following, but rather than considering a pair of parameters $k_{u,v}$ and $r_{u,v}$ for each pair of nodes $(u, v)$ (which implies $2 \times |\mathcal{U}| \times (|\mathcal{U}| - 1)$ parameters to store), we propose to consider neural functions which output the corresponding parameters according to the hidden state of the emitter $u$, depending on its ancestor branch in the cascade, and a continuous embedding of the receiver $v$.

## 3 RECURRENT NEURAL DIFFUSION MODEL

This section first presents our recurrent generative model of cascades. Then, it details the proposed learning algorithm.

### 3.1 RECURRENT DIFFUSION MODEL

As discussed above, we consider that each infected node $v$ in an episode $D$ owns a state $z_v^D \in \mathbb{R}^d$ depending on the path the content followed to reach $v$ in $D$, with $d$ the dimension of the representation space. Knowing the state $z_u^D$ of the node $u$ that first infected $v$, the state $z_v^D$ is computed as:

$$z_v^D = f_\phi(z_u^D, \omega_v^{(f)}) \tag{2}$$

with $f_\phi : \mathbb{R}^d \times \mathbb{R}^d \to \mathbb{R}^d$ a function, with parameters $\phi$, that transforms the state of $u$ according to a shared representation $\omega_v^{(f)} \in \mathbb{R}^d$ of the node $v$. This function can either be an Elman RNN cell, a multi-layer perceptron (MLP) or a Gated Recurrent Unit (GRU). An LSTM could also be used here, but $z_v^D$ should include both the cell and the state of $v$ in that case.

Given a state $z$ for an infected node $u$ in $D$, the probability that $u$ succeeds in transmitting the diffused content to $v$ is given by:

$$k_{u,v}(z) = \sigma\left(< z, \omega_v^{(k)} >\right) \tag{3}$$

where $\sigma(.)$ stands for the sigmoid function and $\omega_v^{(k)}$ an embedding of size $d$ for any node $v \in \mathcal{U}$.

Similarly to the CTIC model, if a node $u$ succeeds in infecting another node $v$, the delay of infection depends on an exponential distribution with parameter $r_{u,v}$. To simplify learning, we assume that the delay of infection does not depend on the history of diffusion, only the probability of infection does. Thus, for a given pair $(u, v)$, the delay parameter is the same for every episode $D$:

$$r_{u,v} = exp(-| < \omega_u^{(r,1)}, \omega_v^{(r,2)} > |) \tag{4}$$

with $|x|$ the absolute value of a real scalar $x$ and $\omega_u^{(r,1)}$ and $\omega_v^{(r,2)}$ correspond to two embeddings of size $d$ for any node $u \in \mathcal{U}$, $\omega^{(r,1)}$ for the source of the transition and $\omega^{(r,2)}$ for its destination in order to enable asymmetric behavior.

We set $\Theta = \left(\phi, z_0, (\omega_u^{(f)})_{u \in \mathcal{U}}, (\omega_u^{(r,1)})_{u \in \mathcal{U}}, (\omega_u^{(r,2)})_{u \in \mathcal{U}}, (\omega_u^{(k)})_{u \in \mathcal{U}}\right)$ as the parameters of our model. The generative process, similar to the one of CTIC, is given in appendix in algorithm 1. In this process, the state of the initial node, the world node $u_0$, is a parameter vector $z_0$ to be learned. The process iterates while there remains some nodes in a set of infectious nodes (initialized with $u_0$). At each iteration, the process selects the infectious node $u$ with minimal time-stamp of infection, removes it from the set of infectious nodes, records it as infected and attempts to infect each non infected node $v$ according to the probability $k_{u,v}(z_u^D)$. If it succeeds, a time $t$ is sampled for $v$ with an exponential law with parameter $r_{u,v}$. If the new time $t$ for $v$ is lower than its current time $t_v^D$ (initialized with $t_v^D = \infty$), this new time is stored in $t_v^D$, $v$ is added to the set of infectious nodes and its new state $z_v^D$ is computed according to its new infector $u$.

### 3.2 LEARNING THE MODEL

As in CTIC, we need to define the probability that the node $u \in U^D$ infects the node $v \in U^D$ at time $t_v^D$ with our model. Given a state $z$ for $u$ in $D$, we have:

$$a_{u,v}^D(z) = k_{u,v}(z)r_{u,v} \exp^{-r_{u,v}(t_v^D - t_u^D)} \tag{5}$$

Also, the probability that $u$ does not infect $v$ before $t_v^D$ given a state $z$ for $u$ in $D$ is:

$$b_{u,v}^D(z) = 1 - k_{u,v}(z) \int_{t_u^D}^{t_v^D} r_{u,v} \exp^{-r_{u,v}(t-t_u^D)} dt = k_{u,v}(z) \exp^{-r_{u,v}(t_v^D - t_u^D)} + 1 - k_{u,v}(z) \quad (6)$$

The probability density that node $v$ is infected at time $t_v^D$ given a set of states $z$ for all nodes infected before $v$ is:

$$h_v^D(\boldsymbol{z}) = \sum_{u \in \mathcal{U}, t_u^D < t_v^D} a_{u,v}^D(z_u) \prod_{x \in \mathcal{U} \setminus \{u\}, t_x^D < t_v^D} b_{x,v}^D(z_x) \quad (7)$$

$$= \prod_{x \in \mathcal{U}, t_x^D < t_v^D} b_{x,v}^D(z_x) \sum_{u \in \mathcal{U}, t_u^D < t_v^D} a_{u,v}^D(z_u) / b_{u,v}^D(z_u) \quad (8)$$

where $z_u$ is the state of node $u$ in $\boldsymbol{z}$. Similarly, the probability density that node $v$ is not infected in $D$ at the end of observation time $T$ given a set of states $\boldsymbol{z}$ for all nodes infected in $D$ is:

$$g_v^D(\boldsymbol{z}) = \prod_{u \in U^D} (k_{u,v}(z_u) \exp^{-r_{u,v}(T-t_u^D)} + 1 - k_{u,v}(z_u)) \approx \prod_{u \in U^D} (1 - k_{u,v}(z_u)) \quad (9)$$

where the approximation is done assuming a sufficiently long observation period.

The learning process of our model is based on a likelihood maximization, similarly to maximizing eq.1 in the classical CTIC model. However, in our case the infection probabilities depend on hidden states $\boldsymbol{z}$ associated to the infected nodes. Since observations only contain infection time-stamps, this requires to marginalize over every possible sequence of ancestors $I$ for every $D \in \mathcal{D}$:

$$\log p(\mathcal{D}) = \sum_{D \in \mathcal{D}} \log p(D) = \sum_{D \in \mathcal{D}} \log \sum_{I \in \mathcal{I}^D} p(D, I) \quad (10)$$

where $\mathcal{I}^D$ is the set $\{v \in \mathbb{N}^{|D|-1} | v_0 = 0 \wedge \forall i > 0, v_i < i\}$ of all possible ancestors sequences for $D$. $p(D, I)$ corresponds to the joint probability of the episode $D$ and an ancestor sequence $I \in \mathcal{I}^D$. Taking $p(D, I) = p(I)p(D|I)$ would lead to an intractable computation of $p(D|I)$ using our recurrent cascade model, since it would imply to estimate the probability of any infection in $D$ according to the full ancestors sequence. Fortunately, using the bayesian chain rule, the joint probability can be written as:

$$p(D, I) = \prod_{i=1}^{|D|-1} p(D_i | D_{<i}, I_{<i}) \, p(I_i | D_{\leq i}, I_{<i}) \prod_{v \notin U^D} p(v \notin U^D | D_{\leq |D|-1}, I) \quad (11)$$

where $D_{<i} = (D_j)_{j \in \{0,...,i-1\}}$ corresponds to the sequence of the $i$ first components of $D$ (the $i$ first components in $U^D$ with their associated time-stamps) and $I_{<i} = (I_j)_{j \in \{0,...,i-1\}}$ stands for the vector containing the $i$ first components of $I$. We have for every $i \in \{1, ..., |D| - 1\}$: $p(D_i | D_{<i}, I_{<i}) = h_{U_i^D}^D(\boldsymbol{z}_{<i}^D)$, where $\boldsymbol{z}_{<i}^D$ is a set containing the states of the $i$ first infected nodes in $D$, which can be deduced from $D_{<i}$ and $I_{<i}$ using the equation 2. We also have: $p(v \notin U^D | D_{\leq |D|-1}, I) = g_v^D(\boldsymbol{z}_{\leq i}^D)$. The probability $p(I_i | D_{\leq i}, I_{<i})$ is the conditional probability that $I(U_i^D)$ was the node who first infected $U_i^D$, given all the previous infection events and the fact that $U_i^D$ was infected at $t_{U_i^D}^D$ by one of the previously infected nodes in $D$. It can be obtained, according to formula 8, via:

$$p(I_i | D_{\leq i}, I_{<i}) = \frac{a_{I(U_i^D), U_i^D}^D \left( z_{I(U_i^D)}^D \right) / b_{I(U_i^D), U_i^D}^D \left( z_{I(U_i^D)}^D \right)}{\sum_{u \in \mathcal{U}, t_u^D < t_{U_i^D}^D} a_{u, U_i^D}^D (z_u^D) / b_{u, U_i^D}^D (z_u^D)} \quad (12)$$

with $I(U_i^D) \in U^D$ the infector of $U_i^D$ stored in $I$.

Unfortunately the log-likelihood from formula 10 is still particularly difficult to optimize directly since it requires to consider every possible vector $I \in \mathcal{I}^D$ for each training episode $D$ at each optimization iteration. Moreover, the probability products in formula 11 would lead to zero gradients because of decimal representation limits. Therefore, we need to define an approach where the optimization can be done via trajectory sampling. Different choices would be possible. First, MCMC

approaches such as the Gibbs Sampling EM could be used, but they require to sample from the posteriors of the full trajectories of the cascades, which is very unstable and complex to perform. The full computation of the posterior distributions could be avoided by using simpler propositional distributions (such as done for instance via importance sampling with auxiliary variables in (Farajtabar et al., 2015) for diffusion source detection), but this would face a very high variance in our case. Another possibility is to adopt a variational approach (Kingma & Welling, 2013), where an auxiliary distribution $q$ is learned for the inference of the latent variables. As done in (Krishnan et al., 2016) for the inference in sequences, a smoothing strategy could be developed by relying on a bi-directional RNN that would consider past and future infections for the inference of the ancestors of nodes via $q(I_i|D, I_{<i})$ for every infected node $i$ in an episode $D$. However, learning the parameters of such a distribution is particularly difficult (episodes of different lengths, cascades considered as sequences, etc.). Also, another possibility for smoothing would be to define an independent distribution $q_i^D$ for every episode $D \in \mathcal{D}$ and every infection $i \in \{1, ..., |D| - 1\}$. However, this induces a huge number of variational parameters, increasing with the size of the training set (linearly in the number of training episodes and quadratically in the size of the episodes). Thus, we propose to rather rely on the conditional distribution of ancestors given the past for sampling (i.e, $q_i^D(I_i) = p(I_i|D_{\leq i}, I_{<i})$), which corresponds to a filtering inference process.

From the Jensen inequality on concave functions, we get for a given episode $D$:

$$
\begin{aligned}
\log p(D) &= \log \sum_{I \in \mathcal{I}^D} p(D, I) \\
&\geq \mathbb{E}_{I \sim q^D} \left[ \log p(D, I) - \log q^D(I) \right] \\
&= \mathbb{E}_{I \sim q^D} \left[ \sum_{i=1}^{|D|-1} \log p(D_i|D_{<i}, I_{<i}) + \sum_{v \notin U^D} \log p(v \notin U^D|D_{\leq |D|-1}, I) \right] \\
&\triangleq \mathcal{L}(D; \Theta)
\end{aligned}
\tag{13}
$$

where $q^D = \prod_{i=1}^{|D|-1} p(I_i|D_{\leq i}, I_{<i})$. This leads to a lower-bound of the log-likelihood, which corresponds to an expectation from which it is easy to sample: at each new infection of a node $i$ in a episode $D$, we can sample from a distribution depending on the past only. Maximizing this lower-bound (also called the ELBO) encourages the process to choose trajectories that explain the best the observed episode. To maximize it via stochastic optimization, we refer to the score function estimator (Ranganath et al., 2014), which leverages the derivative of the log-function ($\nabla_\theta \log p(\mathbf{x}; \theta) = \frac{\nabla_\theta p(\mathbf{x};\theta)}{p(\mathbf{x};\theta)}$) to express the gradient as an expectation from which we can sample. Another possibility would have been to rely on the Gumbel-Softmax and the Concrete distribution with reparametrization such as done in (Maddison et al., 2016), but we observed greatly better results using the log-trick. The gradient of the ELBO function for all the episodes is given by:

$$
\nabla_\Theta \mathcal{L}(\mathcal{D}; \Theta) = \sum_{D \in \mathcal{D}} \mathbb{E}_{I \sim q^D} \left[ \left( \log p^I(D) - b \right) \nabla_\Theta \log q^D(I) + \nabla_\Theta \log p^I(D) \right]
\tag{14}
$$

where $p^I(D)$ is a shortcut for $\prod_{i=1}^{|D|-1} p(D_i|D_{<i}, I_{<i}) \prod_{v \notin U^D} p(v \notin U^D|D_{\leq |D|-1}, I)$ and $b$ is a moving-average baseline of the ELBO per training episode, used to reduce the variance (taken over the 100 previous training epochs in our experiments). This stochastic gradient formulation enables to obtain unbiased steepest ascent directions despite the need to sample the ancestor vectors for the computation of the node states (with the replacement of expectations by averages over $K$ samples for each episode). It contains two terms: while the first one encourages high conditional probabilities for ancestors that maximize the likelihood of the full episodes, the second one leads to improve the likelihood of the observed infections regarding the past of the sampled diffusion path.

The optimization is done using the ADAM optimizer (Kingma & Ba, 2014) over mini-batches of $M$ episodes ordered by length to avoid padding ($M = 512$ and $K = 1$ in our experiments). Our full efficient algorithm is given in appendix [2].

---

[2]The code is publicly available at XXX Anonymous.

## 4 EXPERIMENTS

### 4.1 SETUP

We perform experiments on two artificial and three real-world datasets:

- Arti1: Episodes generated on a scale-free random graph of 100 nodes. The generation process follows the CTIC model. But rather than only one transmission probability $k$ parameter per edge, we set 5 different $k_i$ depending on the diffusion nature. Before each simulation a number $i \in \{1, ..., 5\}$ is sampled, which determines the parameters to use. 10000 episodes for training, 5000 for validation, 5000 for testing. Mean length of the episodes=7.55 (stdev=5.51);

- Arti2: Episodes sampled on the same graph as Arti1, also with CTIC but where each $k_{u,v}$ is a function of the transmitted content and the features of the receiver $v$. A content $z \in \mathbb{R}^5$ is sampled from a Dirichlet with parameter $\alpha = 0.1$ before each simulation and the sigmoid of the dot product between this content and the edge features determines the transmission probabilities. Features of the hub nodes (nodes with a degree greater than 30) are sampled from a Dirichet with $\alpha = 10$ (multi-content nodes), while those of other nodes are sampled from a Dirichet with $\alpha = 0.1$ (content-specific nodes). 10000 episodes for training, 5000 for validation, 5000 for testing. Mean length of the episodes=6.89 (stdev=7.7).

- Digg: Data collected from the Digg stream API during one month. Infections are the "diggs" posted by users on links published on the media. We kept the 100 most active users from the collected data. 20000 episodes for training, 5000 for validation, 5000 for testing. Mean length of the episodes=4.26 (stdev=9,26).

- Weibo: Retweet cascades extracted from the Weibo microbloging website using the procedure described in (Leskovec et al., 2009). The dataset was collected by (Fu et al., 2013). 4000 nodes, 45000 episodes for training, 5000 for validation, 5000 for testing. Mean length of episodes=4.58 (stdev=2.15).

- Memetracker: The memetracker dataset described in (Leskovec et al., 2009) contains millions of blog posts and news articles. Each website or blog stands as a user, and we use the phrase clusters extracted by Leskovec et al. (2009) to trace the flow of information. 500 nodes, 250000 for training, 5000 for validation, 5000 for testing. Mean length of episodes=8.68 (stdev=11.45).

We compare our model recCTIC to the following temporal diffusion baselines:

- CTIC: the Continuous-Time Independent Cascade model in its original version (Saito et al., 2009);

- RNN: the Recurrent Temporal Point Process model from (Du et al., 2016) where episodes are considered as sequences that can be treated with a classical RNN outputting at each step the probability distributions of the next infected node and its time-stamp;

- CYAN: Similar to RMTPP but with an attention mechanism to select previous states (Wang et al., 2017b);

- CYAN-cov: The same as Cyan but with a more sophisticated attention mechanism using an history of attention states, to give more weights to important nodes;

- DAN: the attention model described in (Wang et al., 2018). It is very similar to CYAN but uses a pooling mechanism rather than a recurrent network to consider the past in the predictions. In the version of (Wang et al., 2018), the model only predicts the next infected node at each step, not its time of infection. To enable a comparison with the other models, we extended it by adding a time prediction mechanism similar to the temporal head of CYAN.

- EmbCTIC: a version of our model where the node state $z$ is replaced in the diffusion probability computation (eq. 3) by a static embedding for the source (similarly to the formulation of the delay parameter in eq. 4). This corresponds to an embedded version of CTIC, similarly to the embedded version of DAIC from (Bourigault et al., 2016).

Note that to adapt baselines based on RNN for diffusion modeling and render them comparable to cascade-based ones, we add a "end node" at the end of each episode before training and testing them. In such a way, these models are able to model the end of the episodes by predicting this end node as the next event (no time-delay prediction for this node however).

Our model and the baselines were tuned by a grid search process on a validation set for each dataset (although the best hyper-parameters obtained for Arti1 remained near optimal for the other ones). For every model with an embedding space (i.e., all except CTIC), we set its dimension to $d = 50$ (larger dimensions induce a more difficult convergence of the learning without significant gain in accuracy). The reported results for our model use a GRU module as the recurrent state transformation function $f_\phi$.

We evaluate our models on three distinct tasks:

- Diffusion modelling: the performances of the methods are reported in term of negative log-likelihood of the test episodes (i.e., $NLL = -(1/|\mathcal{D}_{test}|)\sum_{D \in \mathcal{D}_{test}} \log p(D)$). Lower values denote models that are less surprised by test episodes than others, rendering their generalization ability. The NLL measure depends on the model, but for each it renders the probability of an episode to be observed according to the model, both on which nodes are eventually infected and at what time. For our model which has to sample trajectories, the NLL is approximated via importance sampling by considering $p(D) \approx \frac{1}{S}\sum_{s=1}^{S} \frac{p(D,I^{(s)})}{q^D(I^{(s)})}$ computed on $S$ infector vectors sampled from $q^D$. We used $S = 100$ in our experiments;

- Diffusion generation: the models are compared on their ability to simulate realistic cascades. The aim is to predict the marginal probabilities of nodes to be eventually infected. The results are reported in term of Cross-Entropy (CE) taken over the whole set of nodes for each episode: $CE = \frac{1}{|\mathcal{D}_{test}| \times |\mathcal{U}|}\sum_{D \in \mathcal{D}_{test}} \sum_{u \in \mathcal{U}} \log p(u \in U^D)^{\mathbb{I}(u \in U^D)} + \log p(u \notin U^D)^{\mathbb{I}(u \notin U^D)}$, where $\mathbb{I}(.)$ stands for the indicator function returning 1 if its argument is true, 0 else. $p(u \in U^D)$ is estimated via Monte-Carlo simulations (following the generation process of the models and counting the rate of simulations in which $u$ is infected). 1000 simulations are performed for each test episode in our experiments.

- Diffusion Path prediction: the models are assessed on their ability to choose the true infectors in observed diffusion episodes. This is only considered on the artificial dataset for which we have the ground truth on who infected whom. The INF measure corresponds to the expectation of the rate of true infectors chosen by the models: $INF = \frac{1}{\sum_{D \in \mathcal{D}_{test}}(|D|-1)}\sum_{D \in \mathcal{D}_{test}} \sum_{i \in \{1,...,|D|-1\}} p(I_i = inf(i,D)|D_{\leq i})$, with $inf(i,D)$ the true infector of the $i$-th infected node in the episode $D$. For RNN, their is no selection mechanism, it is excluded from the results for this measure. For models with attention (CYAN and DAN), we consider the attention weights as selection probabilities. For cascade based models which explicitly model this, we directly use the corresponding probability $p(I_i = inf(i,D)|D_{\leq i})$. In our model, this corresponds to an expectation over previous infectors in the cascade (i.e., $\frac{1}{S}\sum_{s=1}^{S} p(I_i = inf(i,D)|D_{\leq i}, I_{<i}^{(s)})$), with $I^{(s)}$ the $s$-th sampled vector from $q^D$.

For each task, we report results with different amounts of initial observations from test episodes: infections occurred before a given delay $\tau$ from the start of the episode are given as input to the models, from with they infer internal representations, evaluation measures are computed on the remaining of the episode. In tables 1 to 4, 0 means that nothing was initially observed, the models are not conditioned on the start of the episodes. 1 means that infections at the first time stamp are known beforehand, prediction and modeling results thus concern time-stamps greater than 1 (models are conditioned on diffusion sources). 2 and 3 mean that infections occurred respectively before a delay of $\tau = maxT/10$ and a delay of $\tau = maxT/20$ from the start of the episode are known and used to condition the models. Details on how conditioning our model and the baselines w.r.t. starts of episodes are given in the appendix.

| Arti 1 | | | | |
|---|---|---|---|---|
| NLL | 0 | 1 | 2 | 3 |
| rnn | 25,99 | 23,9 | 16,03 | 11,35 |
| cyan | 27,85 | 25,82 | 17,71 | 12,67 |
| cyan-cov | 26,68 | 24,58 | 16,64 | 11,84 |
| dan | 24,77 | 22,69 | 14,96 | 10,5 |
| ctic | 21,00 | 18,56 | 10,81 | 7,48 |
| embCTIC | 20,96 | 18,53 | 10,78 | 7,49 |
| recCTIC | **19,62** | **14,42** | **9,87** | **6,87** |
| CE | 0 | 1 | 2 | 3 |
| rnn | 0,47 | 0,28 | 0,18 | 0,14 |
| cyan | 0,40 | 0,28 | 0,21 | 0,16 |
| cyan-cov | 0,50 | 0,27 | 0,18 | 0,14 |
| dan | 0,40 | 0,93 | 0,69 | 0,44 |
| ctic | **0,31** | 0,36 | 0,65 | 0,45 |
| embCTIC | 0,47 | 0,35 | 0,25 | 0,18 |
| recCTIC | **0,31** | **0,23** | **0,12** | **0,08** |
| INF | 0 | 1 | 2 | 3 |
| cyan | 0,38 | 0,26 | 0,29 | 0,32 |
| cyan-cov | 0,39 | 0,27 | 0,29 | 0,32 |
| dan | 0,42 | 0,26 | 0,22 | 0,21 |
| ctic | 0,64 | 0,54 | 0,55 | 0,58 |
| embCTIC | 0,62 | 0,51 | 0,49 | 0,51 |
| recCTIC | **0,65** | **0,56** | **0,58** | **0,62** |

| Arti 2 | | | | |
|---|---|---|---|---|
| NLL | 0 | 1 | 2 | 3 |
| rnn | 19,72 | 14,55 | 11,78 | 7,65 |
| cyan | 18,78 | 13,62 | 11,04 | 7,09 |
| cyan-cov | 19,55 | 14,35 | 11,56 | 7,41 |
| dan | 18,71 | 13,49 | 10,86 | 6,84 |
| ctic | 20,08 | 14,26 | 10,18 | 5,99 |
| embCTIC | 19,90 | 14,13 | 10,11 | 5,97 |
| recCTIC | **17,39** | **11,59** | **8,31** | **4,97** |
| CE | 0 | 1 | 2 | 3 |
| rnn | 79,0 | 21,0 | 14,2 | 9,3 |
| cyan | 86,9 | 19,6 | 14,4 | 9,4 |
| cyan-cov | 91,7 | 27,8 | 19,7 | 12,4 |
| dan | 97,9 | 98,6 | 75,7 | 43,8 |
| ctic | **63,1** | 23,2 | 24,7 | 21,8 |
| embCTIC | 65,8 | 22,6 | 17,2 | 12,1 |
| recCTIC | 68,7 | **15,9** | **11,2** | **8,3** |
| INF | 0 | 1 | 2 | 3 |
| cyan | 0,30 | 0,15 | 0,11 | 0,11 |
| cyan-cov | 0,29 | 0,14 | 0,10 | 0,10 |
| dan | 0,42 | 0,31 | 0,23 | 0,20 |
| ctic | 0,73 | 0,67 | 0,61 | 0,58 |
| embCTIC | 0,73 | 0,67 | 0,61 | 0,58 |
| recCTIC | **0,90** | **0,88** | **0,86** | **0,84** |

Table 1: Results on the artificial datasets.

## 4.2 Results

Results on the two artificial datasets are given in table 1. While our approach shows significant improvements over other models for NLL and CE results on the Arti1 dataset (except for CE with weak conditioning), its potential is fully exhibited on the Arti2 dataset, where embedding the history for predicting the future of diffusion looks of great importance. Indeed, in this dataset, there exists some hub nodes through which most of the diffusion episodes transit, whatever the nature of the diffusion. In that case, the path of the diffusion contains very useful information that can be leveraged to predict infections after the hub node: the infection of the hub node is a necessary condition for the infection of its successors, but not sufficient since this node is triggered in various kinds of situations. Depending on who transmitted the content to it, different successors are infected then. Markovian cascade models such as CTIC or embCTIC cannot model this since infection probabilities only depend on disjoint past events of infection, not on paths taken by the propagated content. RNN-based models are better armed for this, but their performances are undermined by their way of aggregating past information. Attention mechanisms of CYAN and DAN attempt to overcome this, but it looks quite unstable with errors accumulating through time. Our approach appears as an effective compromise between both worlds, by embedding past as RNN approaches, while maintaining the bayesian cascade structures of the data. Its results on the INF measure are very great compared to the other approaches. This is especially true on the Arti2 dataset, which highlights its very good ability for uncovering the dynamics of diffusion, even for such complex problems with strong entanglement between diffusions of different natures.

The good behavior of the approach is not only observed on the artificial datasets, which have been generated by the cascade model of CTIC on a graph of relationships, but also on real-world datasets. Tables 2 to 4 report results on the three real-world datasets. In these tables, we observe that RNN based approaches have more difficulties to model test episodes than cascade based ones. The attention mechanism of the CYAN and DAN approaches allow them to get sometimes closer to the cascade based results (especially on Digg), but their performances are very variable from a dataset to another. These methods are good for the task they were initially designed - predicting the directly next infection (this had been observed in our experimentations)-, but not for modeling or long term prediction purposes. This is a strong limitation since the directly next infection does not help much to understand the dynamics and to predict the future of a diffusion. Our approach obtains better results than all other methods in most of settings, especially for the dynamics modelling task (NLL), though infection prediction results (CE) are also usually good compared with its competitors. In-

| NLL | 0 | 1 | 2 | 3 |
|---|---|---|---|---|
| rnn | 31,03 | 26,18 | 24,15 | 23,02 |
| cyan | 16,82 | 11,56 | 9,32 | 8,03 |
| cyan-cov | 21,36 | 16,83 | 14,50 | 13,31 |
| dan | 18,47 | 13,52 | 11,43 | 10,29 |
| ctic | 27,70 | 22,07 | 19,20 | 17,74 |
| embCTIC | 15,98 | 10,31 | 7,92 | 6,75 |
| recCTIC | **15,67** | **10,30** | **7,86** | **6,74** |

| CE | 0 | 1 | 2 | 3 |
|---|---|---|---|---|
| rnn | 0,46 | 0,28 | 0,24 | 0,22 |
| cyan | 0,43 | 0,25 | 0,21 | 0,19 |
| cyan-cov | 0,43 | **0,23** | 0,20 | 0,17 |
| dan | 0,44 | 0,25 | 0,22 | 0,19 |
| ctic | 0,45 | 0,31 | 0,20 | 0,16 |
| embCTIC | 0,43 | 0,31 | 0,21 | 0,17 |
| recCTIC | **0,43** | 0,27 | **0,17** | **0,14** |

Table 2: Negative Log-Likelihood (NLL) and Cross Entropy of Infections (CE) on Digg.

| NLL | 0 | 1 | 2 | 3 |
|---|---|---|---|---|
| rnn | 27,58 | 28,98 | 18,62 | 17,15 |
| cyan | 29,59 | 30,04 | 30,04 | 18,79 |
| cyan-cov | 27,50 | 29,12 | 29,12 | 18,55 |
| dan | 32,35 | 25,02 | 21,97 | 20,39 |
| ctic | 23,92 | 17,88 | 13,31 | 12,28 |
| embCTIC | 24,71 | 18,02 | 13,58 | 12,39 |
| recCTIC | **21,72** | **14,08** | **11,29** | **10,34** |

| CE | 0 | 1 | 2 | 3 |
|---|---|---|---|---|
| rnn | 0,59 | 0,37 | 0,30 | 0,28 |
| cyan | 0,59 | 0,37 | 0,31 | 0,29 |
| cyan-cov | 0,59 | 0,36 | 0,30 | 0,28 |
| dan | **0,58** | 0,26 | 0,26 | 0,24 |
| ctic | **0,58** | 0,25 | 0,31 | 0,28 |
| embCTIC | 0,59 | 0,26 | 0,30 | 0,28 |
| recCTIC | 0,59 | **0,24** | **0,21** | **0,20** |

Table 3: Negative Log-Likelihood (NLL) and Cross Entropy of Infections (CE) on Weibo.

terestingly, while embCTIC usually beats CTIC, recCTIC often obtains even better results. This validates that the history of nodes in the diffusion has a great importance for capturing the main dynamics of the network. Thanks to the black-box inference process and the recurrent mechanism of our proposal, the propositional distribution $q^D$ is encouraged to resemble the conditionnal probability of the full ancestors vector. Regarding the results, the inference process looks to have actually converged toward useful trajectories. This enables the model to adapt distributions regarding the diffusion trajectory during learning. This also allows the model to simulate more consistent cascades regarding sources of diffusion.

## 5 CONCLUSION

We proposed a recurrent cascade-based diffusion modeling approach, which is at the crossroads of cascade-based and RNN approaches. It leverages the best of both worlds with an ability to embed the history of diffusion for prediction while still capturing the tree dependencies underlying the diffusion in network. Results validate the approach both for modeling and prediction tasks.

In this work, we based the sampling of trajectories on a filtering approach where only the past observations are considered for the inference of the infector of a node. Outgoing works concern the development of an inductive variational distribution that rely on whole observed episodes for inference.

| NLL | 0 | 1 | 2 | 3 |
|---|---|---|---|---|
| rnn | 112,3 | 118,2 | 110,4 | 103,6 |
| cyan | 115,2 | 113,1 | 109,2 | 102,1 |
| cyan-cov | 95,58 | 95,05 | 93,64 | 90,20 |
| dan | 91,70 | 89,97 | 86,19 | 78,91 |
| ctic | 52,70 | 55,54 | 48,48 | 44,33 |
| embCTIC | 54,18 | 52,29 | 49,68 | 45,15 |
| recCTIC | **50,11** | **49,34** | **48,35** | **42,20** |

| CE | 0 | 1 | 2 | 3 |
|---|---|---|---|---|
| rnn | 1,68 | 1,66 | 1,59 | 1,51 |
| cyan | 1,66 | 1,64 | 1,59 | 1,49 |
| cyan-cov | 1,61 | 1,59 | 1,52 | 1,39 |
| dan | 1,59 | **1,58** | 1,58 | 1,44 |
| ctic | **1,33** | 1,68 | 1,60 | 1,46 |
| embCTIC | 1,59 | 1,66 | 1,57 | 1,39 |
| recCTIC | 1,51 | 1,60 | **1,49** | **1,36** |

Table 4: Negative Log-Likelihood (NLL) and Cross Entropy of Infections (CE) on Memetracker.

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

# 6 APPENDIX

## 6.1 JOINT PROBABILITY

In this section, we detail the derivation of $p(D, I)$ whose formulation is given in equation 11. For each infected node at position $i$, we need to compute:

- the probability for $U_i^D$ for being infected at its time of infections given the nodes $D_{<i}$ previously infected in $D$ and the states associated to these nodes;

- the probability of the ancestor index $I_i$ given the $i$-th infection $D_i$, and the previous infections $D_{<i}$ associated to their states $z_{<i}^D$;
- the probability that not infected nodes are actually not infected by the $i$-th infected node given its state.

This gives:

$$
\begin{aligned}
p(D, I) = \quad & p(D_0|D_{<0}, z_{<0}^D)p(I_0|D_{\leq 0}, z_{<0}^D) \prod_{v \notin U^D} p(v \notin U^D|D_0, z_0^D) \qquad (15) \\
\times \quad & p(D_1|D_{<1}, z_{<1}^D)p(I_1|D_{\leq 1}, z_{<1}^D) \prod_{v \notin U^D} p(v \notin U^D|D_1, z_1^D) \\
\times \quad & p(D_2|D_{<2}, z_{<2}^D)p(I_2|D_{\leq 2}, z_{<2}^D) \prod_{v \notin U^D} p(v \notin U^D|D_2, z_2^D) \\
& \qquad\qquad\qquad \dots \\
\times \quad & p(D_{|D|-1}|D_{<|D|-1}, z_{<|D|-1}^D)p(I_{|D|-1}|D_{\leq|D|-1}, z_{<|D|-1}^D) \\
& \prod_{v \notin U^D} p(v \notin U^D|D_{|D|-1}, z_{|D|-1}^D) \\
= \quad & \prod_{i=1}^{|D|-1} p(D_i|D_{<i}, I_{<i}) \, p(I_i|D_{\leq i}, I_{<i}) \prod_{v \notin U^D} p(v \notin U^D|D_{\leq|D|-1}, I)
\end{aligned}
$$

## 6.2 GENERATION PROCESS

The generation process of our model is given in algorithm 1. The process iterates while there remains some nodes in a set of infectious nodes (initialized with $u_0$). $\oplus$ denotes the concatenation between two lists. At each iteration, the process selects the infectious node $u$ with minimal time-stamp of infection (all time-stamps but $t_{u_0}^D$ are initialized to $\infty$), removes it from the infectious set and records its infector and infection time-stamp in the cascade. Then, for each node $v$ with time-stamp greater than the one of $u$, $u$ attempts to infects $v$ according to the probability $k_{u,v}(z_u^D)$ (computed with eq 3). If it succeeds, $v$ is inserted in the set of infectious nodes and a time $t$ is sampled for $v$ from an exponential law with parameter $r_{u,v}^D$. If the new time $t$ for $v$ is lower than its stored time $t_v^D$, this new time is stored in $t_v^D$, $u$ is stored as the infector of $v$ in the $from$ table (used to build $I^D$) and the new state $z_v^D$ is computed according to its new infector $u$. The generation process outputs a cascade structure (as described in the introduction of the previous section). From the classical CTIC, the only changes are at lines 12, 14 and 19, respectively for the computation of $k_{u,v}$, $r_{u,v}$ and $z_v^D$.

## 6.3 LEARNING PROCESS

The learning process of our model is depicted in algorithm 2. In this algorithm, the function $makeBins$ first creates minibatches by ordering $\mathcal{D}$ in decreasing length and cutting this ordered list in bins of $batchSize$ episodes each. Each bin contains 3 matrices with $batchSize$ rows (except in the last bin which contains matrices for the remaining $|\mathcal{D}|\%batchSize$ episodes):

- $Inf$: a matrix where the cell $(i, j)$ contains the $j$-infected node in the $i$-th episode of the bin, or $-1$ if the corresponding episode contains less than $j$ infected nodes. The width of the matrix is equal to the number of infected nodes in the longest episode in the bin (the episode in the first row of the matrix);
- $Times$: a matrix where the cell $(i, j)$ contains the infection time-stamp of the $j$-infected node in the $i$-th episode of the bin, or $-1$ if the corresponding episode contains less than $j$ infected nodes. The width of the matrix is equal to the number of infected nodes in the longest episode in the bin (the episode in the first row of the matrix);
- $NotInf$: a binary matrix with $|\mathcal{U}|$ columns where the cell $(i, j)$ equals 0 if the node $j$ is infected in the $i$-the episode of the bin, 1 otherwise;

At each epoch, the algorithm iterates on every bin. For each bin, it first initializes the states of the infected nodes using a function $initStates$ which produces a tensor $\boldsymbol{z}$ of $nbRows(Inf)$ matrices $nbCols(Inf) \times d$ whose each row is filled by $z_0$ (with $nbRows(X)$ and $nbCols(X)$ respectively the number of rows and columns in matrix $X$). For every step $t$ of infection in the bin, the prosess first

---

**Algorithm 1:** Cascade Generation Process

---

1   Input: $\Theta, \mathcal{U}$
2   **for** $u \in \mathcal{U}$ **do**
3     $t_u^D = \infty$
4   **end**
5   $U^D = ()$; $I^D = ()$; $t_{u_0}^D = 0$; $from_{u_0} = 0$; $Infectious = \{u_0\}$;
6   **while** $Infectious \neq \emptyset$ **do**
7     $u \leftarrow \underset{x \in Infectious}{\arg\min} \; t_x^D$;
8     $Infectious \leftarrow Infectious \setminus \{u\}$;
9     $U^D \leftarrow U^D \oplus u$ ;
10    $I^D \leftarrow I^D \oplus from_u$ ;
11    **for** $v \in \mathcal{U} : t_v^D > t_u^D$ **do**
12      $x \sim Bernouilli\,(k_{u,v}(z_u))$;
13      **if** $x == 1$ **then**
14        $x \sim Exp\,(r_{u,v})$;
15        $t \leftarrow t_u^D + t$;
16        **if** $t < t_v^D$ **then**
17          $t_v^D \leftarrow t$;
18          $from_v \leftarrow |U^D| - 1$;
19          $z_v = f_\phi(z_u, \omega_v^{(f)})$;
20          $Infectious \leftarrow Infectious \cup \{v\}$;
21        **end**
22      **end**
23    **end**
24   **end**
25   Output: $C^D = (U^D, (t_u^D)_{u \in \mathcal{U}}, I^D)$;

---

determines in $mask$ the rows of the matrices which correspond to not ended episodes ($Times[:, t]$ refers to the column $t$ of $Times$). Then, if the step is not the initial step $t = 0$, it uses functions $computeLogA$ and $computeLogB$ with nodes previously infected for each episode $Inf[mask, : t]$ associated to their corresponding states $z[mask, : t]$. While the function $computeLogA$ returns a $nbRow(Inf[mask]) \times t$ matrix where the cell $(i, j)$ contains the log-probability for the $j$-th node in the $i$-th episode to infect $Inf[i, t]$ at its infection time-stamp (using a matrix version of equation 5), the function $computeLogB$ returns a same shape matrix where the cell $(i, j)$ contains the log-probability that the $j$-th node in the $i$-th episode does not infect $Inf[i, t]$ before its infection time-stamp (using a matrix version of equation 6).

Then, ancestors at step $t$ are sampled from categorial distributions parameterized by $P(I_t | D_{\leq t}, I_{<t})$ (deduced from logits $A - B$). From them, we compute the log-probability for each infected at step $t$ to be actually infected at their time-stamp of infection by its corresponding sampled infector. (line 19, where $sum(X, 1)$ is a function which returns the vector of the sums of each row from $X$). This quantity is added to the accumulator $ll$.

Line 22 then computes the states for the nodes infected at step $t$ according to the states of the sampled ancestors in $u$ (via the function $computeStates$ which is a matrix version of equation 2).

At the end of each iteration $t$, the log-likelihood that not infected nodes in $NotInf[mask]$ are actually not infected by infected nodes at step $t$ is computed via $computeLogG$, which is a matrix version of equation 9. This quantity is added to the accumulator $ll$.

At the end of the bin (when $t == nbCols$), a control variate baseline is computed by maintaining a list $bh$ of the quantity vectors considered in $\nabla_\Theta \mathcal{L}(\mathcal{D})$. The baseline $b$ considered in the stochastic gradient for any episode $D$ is thus equal to the average of $(logp(D) - 1)$ for this specific episode taken over the $b_l ength$ previous epochs.

Finally, the gradients are computed and the optimizer ADAM is used to update the parameters of the model. Note that this algorithm does not use the gradient update given in eq. 14. It is based on $P(I_t|D_{\leq t}, I_{<t})$ and $P(D_t, I_t|D_{<t}, I_{<t})$ for every $t \in \{0, ..., nbCols(Inf)\}$ (rather than based on the simplification $P(D_t|D_{<t}, I_{<t})$ as given in eq. 13). This is equivalent but greatly more efficient since in both cases $P(I_t|D_{\leq t}, I_{<t})$ needs to be estimated for sampling and $P(D_t, I_t|D_{<t}, I_{<t})$ is much easier to compute than $P(D_t|D_{<t}, I_{<t})$ ($P(D_t, I_t|D_{<t}, I_{<t})$ involves a simple product while $P(D_t|D_{<t}, I_{<t})$ involves a sum of products).

---

**Algorithm 2:** Learning Process

---

1  Input: $\mathcal{D}, \mathcal{U}, batchSize, nbEpochs, \Theta, b\_length$
2  $bins \leftarrow makeBins(\mathcal{D}, \mathcal{U}, batchSize)$;
3  **for** $epoch \in \{1, ..., nbEpochs\}$ **do**
4  $\quad$ $ibin \leftarrow 0$;
5  $\quad$ **for** *(Inf,Times,NotInf)* in bins **do**
6  $\quad\quad$ $ll \leftarrow (0)_{nbRows(Inf)}; logq \leftarrow (0)_{nbRows(Inf)}$;
7  $\quad\quad$ $\boldsymbol{z} \leftarrow initStates(z_0, nbRows(Inf), nbCols(Inf))$;
8  $\quad\quad$ **for** $t \in \{0, ..., nbCols(Inf)\}$ **do**
9  $\quad\quad\quad$ $mask \leftarrow (Times[:, t] >= 0)$;
10 $\quad\quad\quad$ **if** $t > 0$ **then**
11 $\quad\quad\quad\quad$ $A \leftarrow computeLogA(\boldsymbol{z}[mask, : t], Inf[mask, : t], Inf[mask, t])$;
12 $\quad\quad\quad\quad$ $B \leftarrow computeLogB(\boldsymbol{z}[mask, : t], Inf[mask, : t], Inf[mask, t])$;
13
14 $\quad\quad\quad\quad$ # Sample from $P(I_t|D_{\leq t}, I_{<t})$
15 $\quad\quad\quad\quad$ $\boldsymbol{u} \sim Categorical(logits = (A - B))$;
16 $\quad\quad\quad\quad$ $logq[mask] \leftarrow logq[mask] + logPi[mask, \boldsymbol{u}]$
17
18 $\quad\quad\quad\quad$ # Compute $P(D_t, I_t|D_{<t}, I_{<t})$
19 $\quad\quad\quad\quad$ $H \leftarrow A[mask, \boldsymbol{u}] - B[mask, \boldsymbol{u}] + sum(B, 1)$;
20 $\quad\quad\quad\quad$ $ll[mask] \leftarrow ll[mask] + H$;
21
22 $\quad\quad\quad\quad$ $\boldsymbol{z}[mask, t] \leftarrow computeStates(\boldsymbol{u}, \boldsymbol{z}[mask, : t], Inf[mask, : t], Inf[mask, t])$;
23 $\quad\quad\quad$ **end**
24 $\quad\quad\quad$ $ll[mask] \leftarrow ll[mask] + computeLogG(\boldsymbol{z}[mask, t], Inf[mask, t], NotInf[mask])$;
25 $\quad\quad$ **end**
26 $\quad\quad$ $bh[ibin] \leftarrow h[ibin] \oplus (ll - logq - 1)$;
27 $\quad\quad$ **if** $epoch \geq b\_length$ **then**
28 $\quad\quad\quad$ $bh[ibin].pop(0)$;
29 $\quad\quad$ **end**
30 $\quad\quad$ $b \leftarrow sum(bh[ibin], 1)/min(epoch + 1, b\_length)$;
31 $\quad\quad$ $\nabla_\Theta \mathcal{L}(\mathcal{D}; \Theta) \leftarrow \dfrac{1}{nbRows(Inf)} [sum((ll - logq - 1 - b) \nabla_\Theta logq + \nabla_\Theta \log ll)]$;
32
33 $\quad\quad$ $\Theta \leftarrow ADAM(\nabla_\Theta \mathcal{L}(\mathcal{D}; \Theta))$;
34
35 $\quad\quad$ $ibin \leftarrow ibin + 1$;
36 $\quad$ **end**
37 **end**

---

## 6.4 CONDITIONED MODELS

Our experiments in tables 1 to 4 include results obtained with known starts of episodes (columns 1, 2 and 3, for which $\tau > 1$). For these cases, one need to be able to condition the models according to right-censored episodes, which we note $D^\tau$ in the following (every infected node in $u \in U^D$ with $t_u^D \geq \tau$ is reported as not infected in $D^\tau$). For the NLL measure, one need to be able to compute the negative log-likelihood of the end of any episode $D$ given the beginning $D^\tau$ (which means estimating $p(D|D^\tau)$). For the CE measure, one need to estimate the probabilities of final

infections in any episode $D$ given its beginning $D^\tau$ (i.e., estimating $p(u \in U^D | D^\tau)$ via Monte-Carlo simulations).

For models RNN, CYAN and DAN which have deterministic hidden representations, the conditioning on $D^\tau$ is direct: it suffices to traverse the episode from the start to the last infected node in $D^\tau$ to obtain the full representation of the input. Then, the model can be normally used from it on the remaining of the test episode for modeling or generation purposes (except for the time-stamp of the first next event for which one has to take into account that it cannot happen before $\tau$).

For CTIC and EmbCTIC, the conditioning is also easy since future infections from $\tau$ only depend on time-stamps of infections before $\tau$ without any need of trajectory modelling. For simulation purposes and the CE measure, the algorithm 1 can be applied with the infection time-stamps and the $Infectious$ set initialized according to $D^\tau$. Then, the only difference is that infection delays are sampled from a truncated exponential in order to ensure that new infections can not occur before $\tau$. For the NLL measure, one need to reconsider the computation of the infection probabilities from nodes infected before $\tau$, which must take into account that infections did not happened before $\tau$. For each node $v$ which is not infected in $D^\tau$, its probability $a_{u,v}^D$ of being infected by any infected node $u$ in $D^\tau$ is divided by the probability that $u$ did not succeed in infecting $v$ before $\tau$:

$$a_{u,v}^{D|D^\tau} = \begin{cases} \dfrac{k_{u,v} r_{u,v} \exp^{-r_{u,v}(t_v^D - t_u^D)}}{k_{u,v} \exp^{-r_{u,v}(\tau - t_u^D)} + 1 - k_{u,v}} & \text{if } t_u^D < \tau, \\[12pt] k_{u,v} r_{u,v} \exp^{-r_{u,v}(t_v^D - t_u^D)} & \text{otherwise.} \end{cases} \tag{16}$$

Also, for every $v$ with $t_v^D \geq \tau$ and every $u$ with $t_u^D < t_v^D$, we rewrite the probability $b_{u,v}^D$ that $u$ does not succeed in infecting $v$ at $t_v^D$ as:

$$b_{u,v}^{D|D^\tau} = \begin{cases} \dfrac{k_{u,v} \exp^{-r_{u,v}(t_v^D - t_u^D)} + 1 - k_{u,v}}{k_{u,v} \exp^{-r_{u,v}(\tau - t_u^D)} + 1 - k_{u,v}} & \text{if } t_u^D < \tau \text{ and } t_v^D < \infty, \\[12pt] \dfrac{1 - k_{u,v}}{k_{u,v} \exp^{-r_{u,v}(\tau - t_u^D)} + 1 - k_{u,v}} & \text{if } t_u^D < \tau \text{ and } t_v^D = \infty, \\[12pt] k_{u,v} \exp^{-r_{u,v}(t_v^D - t_u^D)} + 1 - k_{u,v} & \text{otherwise.} \end{cases} \tag{17}$$

These quantities are used to compute the conditionnal likelihood $p(D|D^\tau)$:

$$p(D|D^\tau) = \prod_{v \in U^D, t_v^D \geq \tau} h_v^{D|D^\tau} \prod_{v \notin U^D} g_v^{D|D^\tau} \tag{18}$$

where $g_v^{D|D^\tau} = \prod_{u \in U^D} b_{u,v}^{D|D^\tau}$ and, similarly to eq.8 without the dependency in states $\boldsymbol{z}$ for CTIC, the conditionnal $h_v^{D|D^\tau}$ is given as:

$$h_v^{D|D^\tau} = \prod_{x \in \mathcal{U}, t_x^D < t_v^D} b_{x,v}^{D|D^\tau} \sum_{u \in \mathcal{U}, t_u^D < t_v^D} a_{u,v}^{D|D^\tau} / b_{u,v}^{D|D^\tau} \tag{19}$$

At last, for the RINF measure, the only modification is that the statistic is computed solely on nodes $i \in U^D$ such that $t_i^D \geq \tau$.

For our model, beyond the use of conditionnal probabilities (with similar definitions than those from eq.16, 17 and 19 but with $k$ a function depending on the state $z$ of the emitter such as in the section 3.2), we must consider the conditionnal distribution of the ancestors vector of nodes infected before $\tau$, hereafter noted $I^\tau$, given the input $D^\tau$. As done for learning, $I^\tau$ is sampled from our propositional distribution $q^{D^\tau}(I^\tau) = \prod_{i=1}^{|D^\tau|-1} p(I_i^\tau | D_{\leq i}^\tau, I_{\leq i}^\tau)$ and the measures are simple averages on a set of sampled $I^\tau$. Given $S$ samples of $I^\tau$, the NLL is thus computed as:

$$\begin{aligned} NLL &= -(1/|\mathcal{D}_{test}|) \sum_{D \in \mathcal{D}_{test}} \log \frac{1}{S} \sum_{s=1}^{S} p(D|D^\tau, I^{\tau,(s)}) \end{aligned} \tag{20}$$

$$\begin{aligned} &= -(1/|\mathcal{D}_{test}|) \sum_{D \in \mathcal{D}_{test}} \log \frac{1}{S} \sum_{s=1}^{S} \frac{p(D, I^{(s)} | D^\tau, I^{\tau,(s)})}{q^D(I^{(s)} | I^{\tau,(s)})} \end{aligned} \tag{21}$$

with $q^D(I^{(s)}|I^{\tau,(s)}) = q^D(I^{(s)})/q^{D^\tau}(I^{\tau,(s)})$ and where $p(D, I^{(s)}|D^\tau, I^{\tau,(s)})$ is computed same manner as in section 3.2 using conditionnal versions of probability formulations (as given above for CTIC). The last derivation is obtained according to samples of full ancestor vectors $I$, where $I^{(s)}$ is an ancestor vector starting with $I^{\tau,(s)}$ for the infections before $\tau$. Similarly, the CE measure considers marginal probabilities $p(u \in U^D|D^\tau)$ for every node $u \in \mathcal{U}$ defined as averages on simulations performed given $D^\tau$ and sampled $I^{\tau,(s)}$ as input of the generation algorithm 1. Lastly, the INF measure is evaluated by considering:

$$INF = \frac{1}{\sum_{D \in \mathcal{D}_{test}}(|D| - |D^\tau|)} \sum_{D \in \mathcal{D}_{test}} \sum_{\substack{i \in \{1, \dots, |D|-1\}, \\ t_{U_i^D} \geq \tau}} p(I_i = fr(i, D)|D_{\leq i}, I_{<i}^{(s)}) \tag{22}$$

where $I^{(s)}$ is an ancestor vector sampled from $q^D$.

