# OpenReview forum: "A   RECURRENT NEURAL CASCADE-BASED MODEL FOR CONTINUOUS-TIME DIFFUSION PROCESS"
_ICLR.cc/2019/Conference_

### Official Review · AnonReviewer3 · 2018-10-29
**Incremental, writing could be much improved**

**Rating:** 4
**Confidence:** 4

**Review:**

The paper proposes a generative infection cascade model based on latent vector representations. The main idea is to use all possible paths of infections in the model.

Comments:
- The papers clarity could be much improved. It is not easy to follow, is overflowing with notation, and lengthy. Sec. 2.1 for example can easily be made much more concise. Secs. 3.1 and 3.2 are especially confusing. In the first equation in Sec. 3, what is \phi with and without sub/superscript? In Eq. (2), what is k - a probability, or an index? And what is the formal definition "infection" and "future" in the description of k stating that it is "the probability that u infects v in the future"?

- The authors mention that the actual infectors in a diffusion process are rarely observed. While this might be true, in many types of data include infection attempts. This should be worthwhile to model - there are many works on reconstructing cascades from partial data.

- The authors note (rightly) the Eq. (9) is hard to solve, and propose a simple lower bound based on (what I think is) a decomposition assumption.  Unless I misunderstood, this undermines the contribution of the structure of past infections. Could the authors please clarify?

- The results mention 5 (tables?), but only 4 are available, of which one appears floating on the last page.

- Why are methods discussed in the introduction (e.g., DeepCas, Wang 2017a,b 2018) not used as baselines?

Minor:
- Wang 2017a and Wang 2017b are not the same Wang
- Several occurrences of empty parentheses - ()
-

---

> ### Author Response · Authors · 2018-11-25
> **Review answer**
>
> Thanks for your valuable comments and feedback. We attempted to much  improve the clarity of our paper by adding further explanations and correcting typos in many places. New experiments have also been added.
>
> R: "The papers clarity..."
> A: We attempted to lighten sec. 2.1 as possible in the revision. In sec. 3, we agree we made an over-usage of the \phi notation. In the revision we clarified it by changing some notations. We also attempted to clarify things by adding a further description of CTIC we build on at the end of section 2. In eq.2, "(k)" as a superscript denotes, as rather classically done in the literature, that the corresponding parameter is the one of function $k$ (it allows us for instance to distinguish w_u^{(f)}, which is an embedding of u for function f, from w_u^{(k)}, which is an embedding of u for function k. Globally, we tried to make things clearer in sections 2 and 3 in various ways.
>
> R: "The authors mention…"
> A: The context of our work is a task where we only get infection timestamps as input for a set of episodes. We do not have any graph of explicit relationships and we do not know who transmitted the content to whom in the episodes. That is true that many works already dealt with this kind of problems. The problem of cascade reconstruction from diffusion episodes is precisely the task addressed by cascade learning models such as IC, NetRate or CTIC. The proposed approach builds on these models to improve reconstruction by considering the full past of the diffusion rather than relying on the markov assumption of existing cascade models. We attempted to insist on this in the revision.
>
> R: "The authors note… "
> A: Sorry but we cannot really see about which decomposition assumption you are talking about. If you refer to the formulation given in eq. 11, this does not use any assumption. It only uses the Bayesian chain rule, without any simplification of dependencies, as detailed further in appendix 1. This is similar to what is done in other cascade models such as IC or CTIC, but extended for our case where infection probabilities does not only depend on the direct infector of nodes but on the whole history of the diffusion trajectory. The structure of past infections is included in the inferred state of nodes. The only approximation is in the classical ELBO derived in eq. 13 and in the fact that expectations are replaced by sampling: rather than directly optimizing log p(D), which is intractable, we consider a lower bound by introducing an auxiliary distribution q from which it is easier to sample trajectories. Ideally, the distribution q would be equal to p(I|D). In that case, the eq. 13 would become an equality rather than an inequality equation, since q would be proportional to P(D,I). However, this is not possible in our case and we thus use the simpler distribution $q=\prod\limits_{i=1}^{|D|-1} p(I_i|D_{\leq i},I_{<i})$, which corresponds to sampling I_i only regarding past infections rather than the full episode. In the  literature on inference in sequences this is referred as a filtering process (rather than smoothing if full episodes were considered for sampling). This does not undermines the structure of past infections since history is contained in I_{<i} for every index $i$, from which one can easily deduce the recurrent states, and we still consider the full likelihood $p(D,I)$ in the objective function. As we also answered to reviewer 2 who suggested some further derivation in eq.12, we developed a bit more this part to clarify this in the new version of the paper (please refer to our answer to reviewer 2 for more details). The new gradient update (which is equivalent to the former one) contains two terms: while the first one encourages high conditional probabilities for ancestors that maximize the likelihood of the full episodes, the second one leads to improve the likelihood of the observed infections regarding the past of the sampled diffusion path.
>
> R: "Why are methods …"
> A: Wang2017b is actually used in our experiments. It is CYAN, for which we considered two versions. DeepCas and Wang2017a use a graph of relations as input, which we do not have in our setting. Assuming that the explicit relations of the network are not always available or not representative of the true communication channels of the network, the task is to discover diffusion relationships from scratch. In our task, the compared models do not use any graph as input. The models could have been run with the complete graph, but results would be very poor since no reinforcement mechanisms of relations is designed in these models. Also, similarly to Wang2018, these models do not output the infection time of the infected users, which is required to be fairly compared to the approaches considered in the paper. However, to complete the evaluation, we added experiments with an extension of Wang2018 where we added a time prediction mechanism similar to the temporal head of CYAN.

---

### Official Review · AnonReviewer1 · 2018-11-05
**Poor presentation**

**Rating:** 4
**Confidence:** 4

**Review:**

The authors of this paper are proposing a neural network approach for learning diffusion dynamics in networks. The authors argue that the main advantage of their framework is that it incorporates the structure of independent cascades into the model which predicts the diffusion process.

The primary difficulty in reviewing this paper is the poor presentation of the paper. There are many typos and mistakes (e.g., the last paragraph of the paper starts with a sentence that does not make any sense), missing references (e.g., there is an empty parenthesis at the end of the second paragraph on the second page) and in at least two cases, there are references to a formula that is not in the manuscript (e.g., reference to formula 15 on line 3 of page 5). This issues makes reviewing this paper very difficult.

In the modeling section, authors use p(I|D) as q^D(.) in the Eq. 12, where p(I|D) is the conditional probability that a particular node infected an observed infected node first. Plugging p(I|D) in Eq. 12 and using decomposition of p(D ,I) used in Eq. 10, we arrive at a formulation which drops all p(I|D) terms. This results in an objective function which only involves infected nodes (and no term associated with the parent node), weighted by likelihood of each node j infecting the node at step i. This should make the training more simplified than what is discussed in Algorithm 2. Beyond this simplification, I am not clear if that is actually intended by the authors.

The experiments demonstrate a superior performance of the proposed model compared to alternative benchmarks. The authors explain how they trained their own model but there is no mention on how they trained benchmark models. However, given that the datasets used in the experiments were not used in the associated benchmark papers, it is necessary for authors to explain how they trained competing models.

Due to several shortcomings of the paper, most important of which is on presentation of the paper, this manuscript requires a significant revision by the authors to reach the necessary standards for publication, moreover it would be helpful to clarify the modeling choices and consequences of these choices more clearly.

---

> ### Author Response · Authors · 2018-11-25
> **Review answer**
>
> Thanks for your valuable comments and feedback. We attempted to much  improve the clarity of our paper by adding further explanations and correcting typos in many places. New experiments have also been added. Please find bellow our answers to your specific remarks.
>
> R: "In the modeling section, authors use p(I|D) as q^D(.) in the Eq. 12, ...  Beyond this simplification, I am not clear if that is actually intended by the authors."
> A: You are totally right, eq.10 simplifies to $\log p(D) \geq  \mathbb{E}_{I \sim  q^D}  \left[ \sum\limits_{i=1}^{|D|-1} \log p(D_i |D_{<i},I_{<i})  + \sum\limits_{v \not\in U^D} \log p(v \not\in U^D| D_{\leq |D|-1}, I)\right]$. We were aware of this but for simplicity and the conciseness of presentation we chose to not give this final derivation. Also, this was because $P(D_i,I_i|D_{<i},I_{<i})$ is much easier to compute (and more efficient) than $P(D_i |D_{<i},I_{<i})$, given that the computation of $P(I_i| D_{\leq i},I_{<i})$ is in both case required for sampling ($P(D_i,I_i|D_{<i},I_{<i})$ involves a simple product while $P(D_i|D_{<i},I_{<i})$ involves a sum of products). But we agree this was not a good choice, since this simplification appears obvious to the reader. We hesitated a lot on this, our decision was taken to present things as closely as possible to our implementation (by the way there was a mistake in the algorithm 2 resulting from this hesitation - the implementation for our experiments was correct however). We agree that this may appear confusing. Particularly since it gives the feeling that parents of infections are not involved anymore in the computation. But there are actually, since there remains $P(I_i|D_{<i},I_{<i})$ terms in the expectation. The process learns parameters that tend to give high probabilities to the most likely parent vectors regarding infections from the episode. In the revision of the paper, we gave the final derivation you suggested, since it is better for comprehension (It also allowed us to discuss more precisely on what is optimized by considering the given gradient update), and we discuss about its implementation in the appendix as an explanation for the algorithm 2 (that only slightly changed to correct the aforementioned mistake). Thanks for this remark that helped us to much improve the paper.
>
> R: "The authors explain how they trained their own model but there is no mention on how they trained benchmark models"
> A: You are totally right, it is missing. Baseline models are trained on the same training set as our model following the methods proposed in their original paper. Our model and the baselines were tuned by a grid search process on a validation set for each dataset (whose size is given in the description of the datasets),  although the best hyper-parameters obtained for Arti1 remained near optimal for the other ones. For every model with an embedding space (i.e., all except CTIC), we set its dimension to $d=50$ (larger dimensions induce a more difficult convergence of the learning without significant gain in accuracy). We added this explanation in the new version of the paper.

---

### Official Review · AnonReviewer2 · 2018-11-06
**This paper proposes a neural network architectures with locally dense and globally sparse connections. Using dense units a population-based evolutionary algorithm is used to find the sparse connections between modules.**

**Rating:** 7
**Confidence:** 4

**Review:**

The problem that the paper tackles is very important and the approach to tackle it id appealing. The idea of regarding the history as a tree looks very promising. However, it’s noteworthy that embedding to a vector could be useful too if the embedding espace is representative of the entire history and the timing of the events.

Using neural network if an interesting choice for capturing the influence probability and its timing.

The authors need to be clear about their contribution. Is the paper only about replacing the traditional parametric functions of influence and probability with  deep neural networks?
The experimental sections look rather mechanical. I would have put some results on the learned embedding. Or some demonstration of the embedded history or probability to intuitively convey the idea and how it works. This could have made the paper much stronger.

It was nice that the paper iterated and reviewed the possible inference and learning ways. There is one more way. Similar to [1] one can use MCMC with importance sampling on auxiliary variables to infer the hidden diffusion given the observed cascades in continuous-time independent cascade model.

The paper can benefit from a proofreading. There are a few typos throughout the paper such as:
Reference is missing in section 2.1
Page 2 paragraph 1: “an neural attention mechanism”

[1] Back to the Past: Source Identification in Diffusion Networks from Partially Observed Cascades, AISTATS 2015

---

> ### Author Response · Authors · 2018-11-25
> **Review answer**
>
> Thanks for your valuable comments and feedback.
>
> R: "The authors need to be clear about their contribution. Is the paper only about replacing the traditional parametric functions of influence and probability with  deep neural networks? "
> A: Yes but not only. We propose to go beyond the classical markov hypothesis of cascade models that states that any infected node owns the same transmission probabilities whatever from whom comes the propagated content. We indeed do this by replacing the traditional parametric functions with  deep neural networks, which enables to consider recurrent latent states for infected nodes. This allows us to embed the past in node states and hence to output different future diffusion distributions regarding the past trajectory of the propagated content, which is our main contribution (a cascade model with neural network was already proposed for instance in (bourigault et al., 2016) but without past inclusion). While existing works on cascade models learn parameters by inferring the direct infector of every infected node (i.e., estimating $P(I_i|D_{\leq i})$), we need to infer the whole past trajectory to compute node states (i.e., considering $P(I_i|D,I_{<i})$), which is greatly more difficult but the proposed learning approach allowed us to efficiently deal with it.
>
> R: "The experimental sections look rather mechanical. I would have put some results on the learned embedding. Or some demonstration of the embedded history or probability to intuitively convey the idea and how it works. This could have made the paper much stronger."
> A: To give more clues about the good behavior of the algorithm, we added results about the accuracy of the sampled trajectories on the artificial datasets (for which we have the ground truth on who infected whom). We report the rate of good infector choices (i.e., the rate of I_i that equal the ground truth) for our approach and the others. Results show that our approach actually performs better infector choices than CTIC which does not consider the history of the diffusion in its infection probabilities. The use of our recurrent architecture helps the process to distinguish some different diffusion contexts from the past. We also added a second artificial dataset to further analyze the behavior of the approaches.
>
> R: "It was nice that the paper iterated and reviewed the possible inference and learning ways. There is one more way. Similar to [1] one can use MCMC with importance sampling on auxiliary variables to infer the hidden diffusion given the observed cascades in continuous-time independent cascade model."
> A: Thanks for the proposal and the reference that we added in the paper.  The full computation of the posterior distributions could indeed be avoided by using an importance sampling MCMC procedure with auxiliary variables
> (such as done in [1] in the context of diffusion source detection), but in our context we think that the increased computation efficiency would be at the cost of a very higher variance in the learning process, due to the strong intrication of latent and observed variables. In [1], the problem is easier: they do not have to perform optimization on the diffusion parameters (since relying on a diffusion model learned a priori), the problem is to sample hidden infection times to estimate likelihoods and then identifying the most probable source of diffusion.
>
> R: "The paper can benefit from a proofreading."
> A: Thanks, we indeed corrected serveral typos like this in the new version of the paper.

---

### Author Response · Authors · 2018-12-04
**Summary of the revision**

To summary, the main changes in the revision are the following:

    - A new evaluation measure on artificial datasets which considers the rate of correct choices of infectors. This highlights the good ability of our method to discover the paths of diffusion;

    - New experiments on an additional artificial dataset with important hub nodes. In this setting, considering the history of diffusion is crucial to predict the future, which is clearly exhibited in the results (we obtain much better results than CTIC in this setting);

    - An additional baseline in the experiments which corresponds to a recent method using attention without RNN (but whose results remain under those of our method);

    - A further derivation of the ELBO, as suggested by reviewer 2, which enables an easier analysis of what is optimized (while remaining equivalent to the former gradient formulation);

    - A global clarification of the model, with notably an improved presentation of the CTIC model that we build on, a simplification of some notations, and a full proofreading of the paper;

    - More details about evaluation metrics and the tuning of the baselines;

    - A fully re-written experimental analysis section;

    - An additional section in appendix to discuss the conditioning of the models according to observed starts of episodes.

Thanks again to all reviewers for their very relevant comments that enabled us to greatly improve our paper.

---

### Meta-Review · Area_Chair1 · 2018-12-13
**Interesting idea, but paper not ready for publication**

**Confidence:** 3
**Recommendation:** Reject

**Metareview:**

This paper introduces a recurrent neural network approach for learning diffusion dynamics in networks. The main advantage is that it embeds the history of diffusion and incorporates the structure of independent cascades for diffusion modeling and prediction. This is an important problem, and the proposed approach is novel and provides some empirical improvements.
However, there is a lack of theoretical analysis, and in particular modeling choices and consequences of these choices should be emphasized more clearly. While there wasn't a consensus, a majority of the reviewers believe the paper is not ready for publication.